# Local air pollution from oil rig emissions observed during the airborne DACCIWA campaign

Vanessa Brocchi[1], Gisèle Krysztofiak[1], Adrien Deroubaix[2], Greta Stratmann[3], Daniel Sauer[3], Hans Schlager[3], Konrad Deetz[4], Guillaume Dayma[5], Claude Robert[1], Stéphane Chevrier[1], Valéry Catoire[1]

[1] Laboratoire de Physique et Chimie de l'Environnement et de l'Espace (LPC2E), CNRS – Université Orléans – CNES, 45071 Orléans cedex 2, France.
[2] LMD and LATMOS, École Polytechnique, Université Paris Saclay, ENS, IPSL Research University; Sorbonne Universités, UPMC Univ Paris 06, CNRS, Palaiseau, France
[3] Institut für Physik der Atmosphäre, Deutsches Zentrum für Luft und Raumfahrt, Oberpfaffenhofen, Germany.
[4] Karlsruhe Institute of Technology, Institute of Meteorology and Climate Research, Karlsruhe, Germany.
[5] Institut de Combustion, Aérothermique, Réactivité et Environnement (ICARE), CNRS, 45071 Orléans cedex 2, France

*Correspondence to*: Gisèle Krysztofiak (gisele.krysztofiak@cnrs-orleans.fr)

**Abstract.** In the framework of the European DACCIWA (Dynamics-Aerosol-Chemistry-Cloud Interactions in West Africa) project, the airborne study APSOWA (Atmospheric Pollution from Shipping and Oil platforms of West Africa) was conducted in July 2016 to study oil rig emissions off the Gulf of Guinea. Two flights in the marine boundary layer were focused on the floating production storage and offloading (FPSO) vessel operating off the coast of Ghana. Those flights present simultaneous sudden increases of $NO_2$ and aerosols concentrations. Unlike what can be found in flaring emission inventories, no increase in $SO_2$ was detected and an increase in CO is observed only during one of the two flights. Using FLEXPART simulations with regional $NO_2$ satellite flaring inventory in forward trajectory mode, our study reproduces the timing of the aircraft $NO_2$ enhancements. Several sensitivity tests on the flux and the injection height are also performed, leading to the conclusion that a lower $NO_2$ flux helps better reproduce the measurements and that the modification of the injection height does not impact significantly the results of the simulations.

## 1 Introduction

Crude oil extraction from offshore platforms brings raw gas mixed with oil to the surface. Gas flaring is used to dispose of this natural gas in cases where the infrastructure to export it does not exist. This process emits a mixture of trace gases like carbon dioxide ($CO_2$), carbon monoxide (CO), sulfur dioxide ($SO_2$), nitrogen oxides ($NO_x$) as well as particulate matter. Its impacts concern both the ecosystems (Nwaugo et al., 2006; Nwankwo and Ogagarue, 2011) and the air quality (Osuji and Avwiri, 2005). The pollutants can be transported into the free troposphere (Fawole et al., 2016b) or can reach coastal cities in the marine boundary layer (MBL). Flaring emissions can be derived from remote sensing techniques (Elvidge et al., 2013) by the widely used Visible Infrared Imaging Radiometer Suite (VIIRS) Nightfire satellite product (Deetz and Vogel, 2017).

Oil and gas extraction activities are however uncertain in terms of emitted quantities (Tuccella et al., 2017) and direct measurements are necessary.

In Africa, most of the studies focused on the environmental impact of Nigerian oil platform emissions (e.g. Anomohanran, 2012; Hassan and Kouhy, 2013; Asuoha and Osu Charles, 2015; Fawole et al., 2016a) as it is part of the five countries with the highest flaring amounts (Elvidge et al., 2015). Other countries in the Gulf of Guinea are nevertheless affected by environmental problems. The DACCIWA (Dynamics-Aerosol-Chemistry-Cloud Interactions in West Africa; Knippertz et al., 2015) project conducted fieldwork in southern West Africa (SWA) in 2016 to investigate the impact of anthropogenic emissions, notably on air quality. One part of the project included airborne measurements of flaring emissions in order to fill the data gap on oil extraction activities in SWA. Those in situ measurements are, as far as we know, the first reported data on oil extraction activities in this region.

We use here the Lagrangian transport model FLEXPART (FLEXible PARTicle dispersion model; Stohl et al., 2005) in combination with an inventory of emissions dedicated to flaring emissions and created in the framework of the DACCIWA project (Deetz and Vogel, 2017) to reproduce the aircraft measurements. We focus on evaluating the sensitivity of the model to two parameters: the emission flux and the injection height of the flaring emissions. After a brief description of the DACCIWA project and the platform in Sect. 2, we present in Sect. 3 the model and the flaring emission inventory we used, as well as the estimation of the flaring plume injection height. We discuss the modeling results in Sect. 4.

## 2 The DACCIWA/APSOWA project

### 2.1 Description of the campaign

The EU-funded project DACCIWA focuses on the coupling between dynamics, aerosols, chemistry and clouds (Flamant et al., 2017). The research campaign was set up in June-July 2016 and undertook activities ranging from airborne measurements to running atmospheric numerical models. A map showing the location of the research site is presented in supplementary material (Fig. S1).

Our Atmospheric Pollution from Shipping and Oil Platforms of West Africa (APSOWA) project complementing DACCIWA seeks to characterize gaseous and particulate pollutants emitted by oil platforms off the coast of the Gulf of Guinea by dedicated flights conducted with the DLR (Deutsches Zentrum für Luft- und Raumfahrt) research aircraft Falcon-20. Different instruments for gas and particulate measurements were deployed onboard. We focus on CO, $NO_2$, $SO_2$ and aerosols measurements during two flights on July 10 and 14. Both CO and $NO_2$ were measured by SPIRIT (SPectromètre InfraRouge In situ Toute altitude, Catoire et al., 2017). This infrared absorption spectrometer uses continuous-wave distributed-feedback room-temperature Quantum Cascade Lasers (QCLs) allowing online scanning of mid-infrared rotational-vibrational lines with spectral resolution of $10^{-3}$ $cm^{-1}$. In the present campaign, the ambient air is sampled in a multipass cell with a pathlength of 134.22 m and in micro-windows around the 2179.772 $cm^{-1}$ line for CO and the 1630.326 $cm^{-1}$ line for $NO_2$. A home-made

software using the HITRAN 2012 database (Rothman et al., 2013) is used to deduce the total molecule abundance. The overall uncertainties are 4 ppbv for CO and 0.5 ppbv +5% for $NO_2$ at 1.6 s time resolution.

$SO_2$ measurements are performed using a pulsed fluorescence $SO_2$ analyser (Thermo Electron, Model 43C Trace Level). Ultraviolet (UV) light is absorbed by $SO_2$ molecules in the sample gas which become excited and subsequently decay to a lower energy state. The emitted light is detected by a photomultiplier tube and is proportional to the $SO_2$ concentration in the sample gas. The time resolution of the measurements is 1 s, with a moving average of 30 s to smooth the data. The lower detection limit is 0.315 ppbv. The instrument was multipoint calibrated before and after the campaign.

Total aerosol concentrations were measured with a butanol-based condensation particle counter (CPC TSI 3010, modified for aircraft use; Schröder & Ström, 1997 and Fiebig et al., 2005). The particle counter was mounted inside the fuselage behind the Falcon isokinetic aerosol inlet. The large particle cutoff diameter imposed by the inlet has been found to be between 1.5 and 3 μm depending on flight altitude (Fiebig, 2001). The lower cutoff diameter of the CPC was at ~10 nm. Particle losses due to diffusion effects were minimized by using a 5.75 std l/min bypass flow to the instrument. Using the particle loss calculator described in von der Weiden et al. (2009) we estimate the particle losses in the tubing to be less than 10% for the relevant size range. Flight sequences inside clouds are known to cause sampling artefacts and have therefore been removed from the data set.

## 2.2 Flight planning and FPSO description

The first Floating Production Storage and Offloading (FPSO) vessels have been installed in Indonesia in 1974. Since this year, their number has steadily increased (Shimamura, 2002). The concept of those platforms based on ship structure makes possible the development of small size oil fields and the exploitation of them further from the coast and thus in deeper water (Shimamura, 2002). Those platforms have other advantages (Shimamura, 2002): they are faster to build than other floating structures, they have inbuilt storage capability and thus do not necessitate pipelines, and they are movable and easily implantable on another oil field. Because of all those reasons, the FPSO systems will continue to develop. Among the techniques used to dispose of the gas associated with the crude oil extraction, one is the flaring, consisting in burning the gas in an open flame through a stack. This leads to a mixture of emitted gases which can reach the free troposphere if the meteorological conditions are favorable (Fawole et al., 2016b). The two targeted flights, on July 10 and July 14, consisted in about 3-4 hours of meandering transects through emission plumes in the MBL about 300 m above sea level (asl) off the coast of West Africa from Ivory Coast to Togo. We focus on the flights in the vicinity of the FPSO Kwame Nkrumah platform, on the Jubilee Field, off the coast of Ghana (4°35'47.04 N, 2°53'21.11 W). It measures 330 m in length by 65 m in width, and its height up to the top of the chimney is estimated to be 112 m asl. During the campaign, no contact with this FPSO could be obtained in order to have more information on its functioning.

## 3 Method

### 3.1 Lagrangian particle dispersion modelling

The Lagrangian model FLEXPART (FLEXible PARTicle; Stohl et al., 2005) is used to study the transport of the emitted plume from the FPSO in the MBL. It simulates long-range transport and dispersion of atmospheric tracers released over time by computing trajectories of a large number of tracer particles. Model calculations are based on meteorological data from the European Centre for Medium-Range Weather Forecasts (ECMWF), ERA-INTERIM L137 (Dee et al., 2011) extracted every 3 hours and with a horizontal grid mesh size of $0.5° \times 0.5°$. The calculations are performed in forward dispersion mode with the model version 9.0. The particles are released with the chemical properties of $NO_2$, CO and $SO_2$ using constant emissions from Deetz and Vogel (2017) inventory during 7 h with a spin-up of 5 h, allowing the model to be balanced independently from the initial conditions. During the simulation, the $NO_2$ and $SO_2$ like particles mass is lost by wet and dry deposition and by OH reaction (concentrations from GEOS-CHEM model; Technical note FLEXPART v8.2, http://flexpart.eu/downloads/26), which allows a lifetime of about 3 h at 298 K in the MBL for $NO_2$. CO like particles mass is only lost by OH reaction.

### 3.2 Flaring emission inventory

Our study is based on a new gas-flaring emission inventory developed for the DACCIWA project (Deetz and Vogel, 2017). This inventory provides emissions of CO, $CO_2$, NO, $NO_2$ and $SO_2$ for June-July 2014 and 2015 and we use a 2016 updated version for the period of the APSOWA-DACCIWA campaign. It is based on remote sensing observations using VIIRS nighttime radiant heat in combination with combustion equations from Ismail and Umukoro (2014). The emission estimation method is described in detail in Deetz and Vogel (2017). Only the assumptions of interest for our study are indicated: first, the natural gas composition includes 0.03% $H_2S$ (Sonibare and Akeredolu, 2004). Second, this composition measured in the Niger Delta is valid for West Africa in general. Third, the source temperature is deduced from the VIIRS measurements on a monthly mean. For the FPSO platform, it is set to 1600 K, which is a good order of magnitude since the flame temperature can be as high as 2000 K (Fawole et al., 2016b). Fourth, for such a temperature, $NO_2$ is considered as a primary pollutant coming from the rapid conversion of NO close to the source, and the inventory does not include any later transformation of the species. The CO, $SO_2$ and $NO_2$ fluxes estimated with this method for our two days of interest for the FPSO platform are $1.1 \times 10^{-1}$, $4.23 \times 10^{-5}$ and $7 \times 10^{-2}$ kg s$^{-1}$, respectively. Note that the highest uncertainties (+33; -79%) associated to the estimation of gas flaring emissions arise from the parameters required in the combustion equations, e.g. the gas composition, the source temperature and the flare characteristics.

### 3.3 Estimation of the flaring plume injection height

The oil flaring emissions are generally emitted at higher temperatures than the temperature of the surrounding environment, which implies an important role for the buoyancy at the stack exit. Indeed, the buoyancy corresponds to a density ratio between the air parcel and its colder surrounding environment and leads to the rise of this parcel under the influence of

gravity. This effect is to be distinguished from the momentum effect defined as the product of an element mass by its velocity, which can be neglected for such high temperature plume (Briggs, 1965). The buoyancy raises the plume above its initial injection height and can lead to a source height that can be several times higher than the real height of the stack (Arya, 1999). The calculation required to determine the rise of a plume depends on the wind conditions and the atmospheric stability. Before determining those parameters, we determine the MBL height that defines in which part of the troposphere we flew. The MBL is about 680 m asl on July 10 and 582 m asl on July 14 according to the European (ECMWF) and US (NCEP) operational forecasts. The wind conditions are determined by the Falcon-20 measurement system. An average of the wind speed measurements is calculated during the flight period in the vicinity of the platform (see Sect. 4.1). For July 10, an 8 minute mean wind speed of 9.4±0.5 m s$^{-1}$ has been calculated, where the standard deviation represents the natural variability, which is larger than the measurement uncertainty. For July 14, a 7 minute wind speed mean of about 6.6±0.7 m s$^{-1}$ has been calculated. Thus, for both days the conditions are windy.

Concerning the potential temperature for July 10 and 14, the atmosphere is considered as stable with a positive potential temperature gradient. With the previous parameters defined, it is possible to calculate the plume injection height $\Delta H$ (in m) by using Eq. (1) reported by Briggs (1965, 1984):

$$\Delta H = 2.6 \left(\frac{F_b}{us}\right)^{1/3} \tag{1}$$

with $F_b$ as the buoyancy effect in m$^4$ s$^{-3}$ as defined in Eq. (2), $u$ as the wind speed (in m s$^{-1}$) and $s$ as the stability parameter in s$^{-2}$ defined in Eq. (3) (Briggs, 1965):

$$F_b = g \frac{\Delta T}{T_s} wr^2 \tag{2}$$

$$s = \frac{g}{T} \left(\frac{\partial T}{\partial z}\right) + \Gamma \tag{3}$$

$T, T_s, w$ and $r$ are the absolute temperature of the ambient air, the absolute temperature of the stack gases, the vertical velocity of the effluent at the stack exit and the radius of the stack, respectively. $\Delta T$ is the difference between the two temperatures. In Eq. (3), $g$ is the gravitational acceleration, $z$ the altitude and $\Gamma$ the adiabatic lapse rate. Both the temperature at the stack exit and the effluent velocity are taken from Deetz and Vogel (2017). The calculated plume injection height $\Delta H$ is 27 m for July 10 and 14. Briggs' algorithm underestimates the plume rise in stable atmospheric condition (Akingunola et al., 2018). Therefore, another method has been tested to determine the injection height, based on VDI 3782 (1985). For a stable atmosphere, the injection height $\Delta H$ (in m) of the plume is determined by Eq. (4):

$$\Delta H = 74.4 \times Q^{0.333} \times u^{-0.333} \tag{4}$$

with $u$ as the wind speed in m s$^{-1}$ and the unit of the coefficient (74.4) in s$^4$ kg$^{-1}$ m$^{-2}$. The heat flow $Q$, in units of MW, (Eq. 5) is defined in Deetz and Vogel (2017) as:

$$Q = H \cdot \frac{1}{f} \tag{5}$$

with $H$ as the radiant heat observed by VIIRS, in units of MW, and $f$ as the fraction of radiated heat set to 0.27 by Deetz and Vogel (2017), after having averaged the $f$ values given in Guigard et al. (2000). The calculated plume injection heights are, according to Eq. (4), 68 m and 77 m for July 10 and 14, respectively.

The two methods of injection height calculation do not give similar results. The control run (CTRL; Table 1) will use Briggs' algorithm with an injection height of the particles of 27 m for both days. A sensitivity test (ST1; Table 1) will be performed to see the impact on the results of the injection height in the model by using the injection height from VDI 3782 (1985).

## 4 Results and discussion

### 4.1 Description of the measurements

Figure 1 presents the part of the flights in the vicinity of the FPSO platform. During the flight on July 10, the flaring plume was crossed several times (Fig. 1a). It led to four $NO_2$ peaks between 12:33 and 12:45 UTC at an altitude of around 300 m (Fig. 1c). Four simultaneous peaks of aerosols were measured (Fig. 1c). No simultaneous CO peaks (within its lower detection limit 0.3 ppbv; Catoire et al., 2017), neither $SO_2$ peaks (within its lower detection limit 0.3 ppbv (see section 2.1) were detected in the plume of the FPSO platform for this flight. No peak was measured during a second series of transect at a higher altitude (around 600 m, Fig. 1a).

During the flight on July 14, three $NO_2$ peaks were measured during the first transects between 10:48 and 11:00 UTC at around 300 m of altitude downwind the FPSO vessel (Fig. 1b). Those peaks were simultaneously measured with aerosol and CO peaks (Fig. 1d), but still no $SO_2$ peak was detected. Knowing that $SO_2$ comes from $H_2S$ combustion (Sonibare and Akeredolu, 2004), these results suggest that a gas composition of 0.03% of $H_2S$ induces an emission of $SO_2$ concentration lower than the detection limit of the instrument from 3 km of our measurements or the natural gas composition given by Deetz and Vogel (2017) for the Niger Delta is different from that in Ghana for those two flights. The presence (or no) of CO peaks is discussed in Sect. 4.3.

Moreover, SPIRIT allows measurements every 1.6 s. Considering the case on July 10 where we have the maximum aircraft speed (118 m s$^{-1}$) and the shortest peak (lasting about 16 s), the plume width is larger than the measurement e-fold time multiplied by the flight speed. Thus, for all the narrow peaks, the maximum plume concentration is real, not a plume diluted with its surrounding environment.

### 4.2 FLEXPART simulations of the flaring emissions

In order to confirm that the peaks detected by the aircraft instruments correspond to the flaring emissions from the platform and to simulate them, forward trajectories are calculated using FLEXPART. First, as a reference, a simulation called "control run" (CTRL) is run, in which the $NO_2$ flux for the FPSO platform is taken from Deetz and Vogel (2017), i.e. $7 \times 10^{-2}$ kg s$^{-1}$, and the injection height is 27 m, calculated with Briggs' method (see Sect. 3.3). A comparison of the wind speed and

direction between ECMWF simulations and the Falcon measurements is presented in supplementary material. The wind speed and direction derived from ECMWF agree within 1 m s$^{-1}$ and ~1.8°, respectively, for July 10 (Fig. S2) and within ~1 m s$^{-1}$ and ~7°, respectively, for July 14 (Fig. S3). So the transport is well reproduced in FLEXPART. Figure 2a compares observed and simulated time series of $NO_2$ concentrations for the flight on July 10. The four measured peaks are all

remarkably well reproduced in time by FLEXPART. Another study by Tuccella et al. (2017), using WRF-Chem, was able to reproduce the simulated peaks simultaneously to the measured ones for oil facilities emissions in the Norwegian Sea. A $NO_2$ background concentration has been added to the last two peaks. This value is an average of the measurements taken outside the plume and is representative of the ambient pollution not taken into account in FLEXPART as this model only simulates the pollution coming from the platform. Concerning the second and the fourth peak (Fig. 2a), the measurements show two

close peaks that FLEXPART cannot simulate individually, leading to a single and broader simulated peak. This is probably due to an error in the dispersion modelling induced by the horizontal and vertical wind field resolution that prevents us from comparing peak-to-peak concentrations. Even with a finer wind field grid mesh of 0.125°×0.125° (simulation not shown) such close peaks cannot be distinguished, suggesting a still insufficient spatial resolution. Instead, the integrated area under each of the measured and simulated plume transects will be compared and presented in Figure 3 with the percentages

representing the relative differences with respect to SPIRIT measurements. Figure 2d also compares the observed and simulated $NO_2$ concentrations for the flight of July 14. The simulation gives three peaks concomitant with the three measured peaks of interest.

Comparisons between CTRL run and SPIRIT measurements for both flights (Fig. 3, panels A-1 for July 10, B-1 for July 14) show that the simulated concentrations are always overestimated for all the peaks (percentage larger than 0%). Sensitivity

tests are performed in order to show the FLEXPART response to the flux. Fluxes lower than 0.07 kg.s$^{-1}$ are tested from 0.035 to 0.05 kg s$^{-1}$. Figure 3 (panels A-1 and B-1) shows the variations in the differences between the measurements and the FLEXPART simulations with the flux used in FLEXPART for the injection height from Briggs (1965). The plots for both days show a linear response of FLEXPART to the flux increase with a best estimation reached for 0.04 kg.s$^{-1}$ for the flight on July 10 and 0.035 kg.s$^{-1}$ for the flight on July 14. Panels A-2 and B-2 of Figure 3 show similar conclusions but for the

injection height from VDI 3782 (1985). To determine whether the observed linear relationship between the percentages and the flux or the injection height occurs by chance, a simple F-test is performed, assuming that the variances are homogeneous and the results follow a Gaussian distribution. F statistic coefficients are calculated and compared to the 95% or 90% confidence interval with (1, N-2) (N: total number of results) as degrees of freedom (see values in brackets [u;+∞[ in Figure 3). If the value F is included in the confidence interval then the relationship can be considered as linear. The standard errors

on the slope are also added in the plots of Figure 3.

For the flight on July 10, the standard error of the slope coefficients and the F-test (95% of confidence) show linear relationships between the percentage difference and the flux. Only the results on July 14 for the plot with the injection height of VDI 3782 (1985, panel B-2) show a positive F-test but with a 90% of confidence. No conclusions can be drawn for the results on July 14 with the injection height of Briggs (1965, panel B-1). In order to show the response of FLEXPART to the

injection height, panels A-3 and B-3 (Fig. 3) show the percentages versus the injection height (Briggs (1965) or VDI 3782 (1985)). FLEXPART shows similar results regardless of the injection height used as input whatever the flux used. All the cases show standard error on the slope coefficients larger than the slope itself and a F- value not included in the confidence interval (at 95% of confidence as shown in the Figure, and even 90%, not shown) . These results suggest that the differences

between the two injection heights are not significant enough with respect to the vertical resolution of the model or the measurements are too far to be influenced by the changes in this parameter. However, to really conclude about the injection height and to evaluate the flux, more measurements are needed at different altitudes and distances from the emission source.

Besides the weather conditions and the functioning of the platform, the flight location is also an important parameter to be able to evaluate our measurements. Figures 2 b and e show the $NO_2$ plume simulated with FLEXPART as a function of

distance from the source and altitude on July 10 and 14, respectively. The aircraft measurements, represented by the colored circles, are located at the upper part of the plume, away from the strongest concentrations. The work carried out is thus limited by the flight trajectories which were too high and too far from the FPSO platform to catch the part of the plume with the highest concentrations. The operational conditions during the flights were complex for the pilots, and safety concerns forced us to respect a minimum flight level (300 m) and a minimum distance from the source. Finally, we found that the $NO_2$

concentration difference between the measurements and the simulations does not seem to depend on the distance from the source since the measurements are already too far.

We can use the best simulation on each day to estimate the percentage of pollutants transported inside and above the MBL. In both cases, about 90% of the pollutants stay inside the MBL and are susceptible to impact the population living along the coastline. Measurements made along the coastline have shown that $NO_2$ concentrations are generally greater than 2 ppb for

suburban sites and greater than 20 ppb near industrial sites (Bahino et al. 2018). Given the wind velocity (from 6.6 to 9.4 m s$^{-1}$), the air masses attain the coast in 2 to 3 hours, which does not allow to bring significant $NO_2$ concentrations to impact air quality in this area. This is confirmed looking at FLEXPART simulations in Fig. 2b and 2e. They show that $NO_2$ concentrations are already very low (< 1 ppbv) from 40 km from the source on July 10, and even closer (from 20 km from the source) on July 14. The distance between the coast and the emission source following the wind direction being 70 km,

only pollutants with a relatively long lifetime or secondary pollutant as $O_3$ can impact the air quality of the coast.

Considering the very low $SO_2$ flux, the FLEXPART simulations in the CTRL run induce insignificant $SO_2$ concentration at aircraft sampled location (results not shown).

We performed CO simulations as CO peaks were measured in one case out of two (Fig. 2c and 2f). For those simulations, we used the injection height of 27 m of the CTRL run and the flux from Deetz and Vogel (2017) of $1.1\times10^{-1}$ kg s$^{-1}$. For the flight

on July 10, FLEXPART simulates four CO peaks concomitant with the four $NO_2$ peaks while no increase in CO has been observed. On July 14, three peaks of CO are simulated at the same time as the three measured ones, but underestimated. CO is always included as a gas emitted from flaring in the inventory for this specific vessel while it does not seem to be the case each time. A discussion on the flaring combustion processes is presented below.

**4.3 Combustion processes involved in oil flare**

From a combustion point of view, flares generate non-premixed highly turbulent flames, characterized by high-frequency fluctuating flow fields. Flares can be air-assisted or steam-assisted in order to achieve a better efficiency. The turbulence increases the mixing and affects the chemical reaction process. Very recent attempts to model such flames (Aboje et al., 2017; Damodara et al. 2017) showed the influence of the composition of the natural gas and the validity of the chemical kinetic mechanism are of primary importance to predict the emitted species. Moreover, the flame characteristics such as temperature, height, and length, have a strong impact on the dispersion of the plume together with wind speed and other meteorological variables (Rahnama et al., 2016). All these parameters need to be taken into account when considering air field campaigns.

McDaniel (1983) clearly demonstrated CO emissions are strongly related to flare efficiency. When the efficiency is ca. 99.8%, CO is below 10 ppmv (Fig. A-5 in McDaniel (1983)) while CO emissions may reach 440 ppmv when the efficiency drops to 64% (Fig. A-8 in McDaniel (1983)). CO emissions can clearly be linked to the quality of the combustion, which is directly impacted by the turbulence. The observed difference between July 10 and 14 in terms of CO emissions mostly lies in the different wind conditions between these two days: first, the wind speed was lower on July 14, which makes less $O_2$ available to burn with natural gas; second, it appears that the wind direction was not clearly established, as can be seen from the much more dispersed plume in Fig. 1b, resulting in incomplete combustion pockets favoring CO formation. However, a decrease in efficiency should also lead to lower temperatures and $NO_x$ emissions, which is not observed here. The results of this campaign would require to be analyzed in the light of computational fluid dynamic simulations, accounting for a realistic natural gas composition and its high-temperature chemistry, which are beyond the scope of this study.

**5 Conclusions**

This study was conducted in the framework of the DACCIWA FP7 European project in July 2016 in southern West Africa. One target of the project was to measure emissions from oil rigs, which were not well estimated until then. With two flights planned in the vicinity of a FPSO platform, we reported the first flaring in-situ measurements in this region. The aim of this work was to evaluate the capacity of FLEXPART model to reproduce the $NO_2$ airborne measurements and to evaluate the inventory of Deetz and Vogel (2017) in the case of point sources of pollution such as oil platforms. The injection height of the plume was estimated by performing different calculations. According to several sensitivity runs, it appears that the emission flux given by Deetz and Vogel (2017) overestimates the concentrations whereas a lower $NO_2$ flux roughly reproduces the measurements. However, we did not know if the FPSO was under standard conditions of functioning. Concerning the injection height, the sensitivity tests are not conclusive, showing the need for more and better-targeted measurements. An estimation of the pollutant distribution above or inside the MBL shows that the pollutants stay mainly inside the MBL, limiting the transport to the coastline located 70 km downwind of the FPSO.

Sources of uncertainties are associated with the different calculations and hypotheses but the work is mainly limited by the flights trajectories, too far and too high from the platform. So it remains necessary to better quantify the emissions released

during the flaring processes, locally but also at wider scales. Generally speaking, this study suggests that for flights planned in the heart of a flaring plume, it should be possible to link the flaring observations obtained by satellites with the emissions deduced from the airborne measurements. If this relationship is possible, a general relationship between the emissions and the radiant heat could thus allow estimating the emissions of all flaring processes detected by satellite.

**Data availability.** The aircraft data used here can be accessed using the DACCIWA database at http://baobab.sedoo.fr/DACCIWA/ (Brissebrat et al., 2017). A two-year embargo period applies after the upload. External users can request the release of datasets before the end of the embargo period.

**Author contributions.** HS was the mission scientist for the Falcon-20 aircraft and VC the PI of the EUFAR2-APSOWA
campaign. VC, SC and VB participated in the SPIRIT measurements onboard the aircraft with remote assistance from CR for the SPIRIT maintenance. VB, GS and DS performed flight data analyses for their respective instruments. KD provided the updated version of his inventory for the period of the campaign and the VDI report. AD performed simulations using WRF-Chem to simulate the FPSO plume and gave advice for the modeling method. GK and VB performed the FLEXPART simulations. GD conducted the combustion part of the paper. VB wrote the manuscript with contribution from all co-authors.

**Acknowledgments.** We thank P. Jacquet for instrumental support before and during the campaign. The DLR crew is acknowledged for flying operations. This work was funded by the EU FP7 EUFAR2 Transnational Access project and DACCIWA project (Grant Agreement N°603502), the Labex VOLTAIRE (ANR-10-LABX-100-01), the PIVOTS project provided by the Région Centre – Val de Loire (ARD 2020 program and CPER 2015–2020) and the APSOWA project from INSU-LEFE-CHAT program. We thank F. Contino (Vrije Universiteit Brussel, Belgium) for the work undertaken on the
simulations of flame. We also thank G. De Coetlogon and A. Weill (Laboratoire Atmosphères Milieux Observations Spatiales, Université Paris-Saclay and CNRS, France) for their help in determining the MBL height.

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

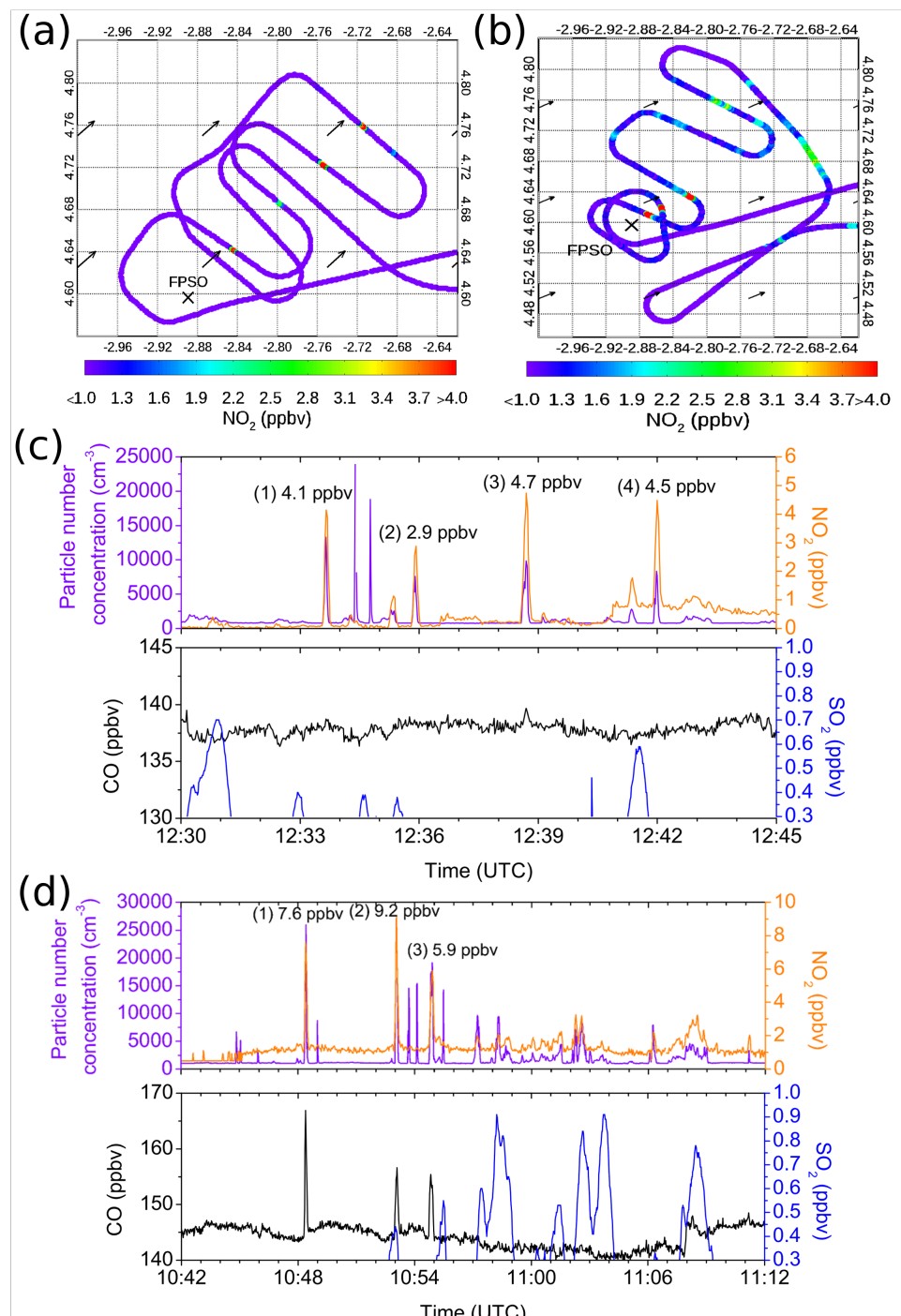

**Figure 1:** (a) NO₂ concentration as a function of the flight trajectory downwind of the FPSO plume for July 10. The black arrows show the wind direction (from ECMWF). (c) NO₂, aerosol, CO and SO₂ concentrations as a function of time zoomed in a part of the flight trajectory in (a). The peaks studied are labelled by a number (from 1 to 4). (b) and (d) are similar to (a) and (c) for July 14.

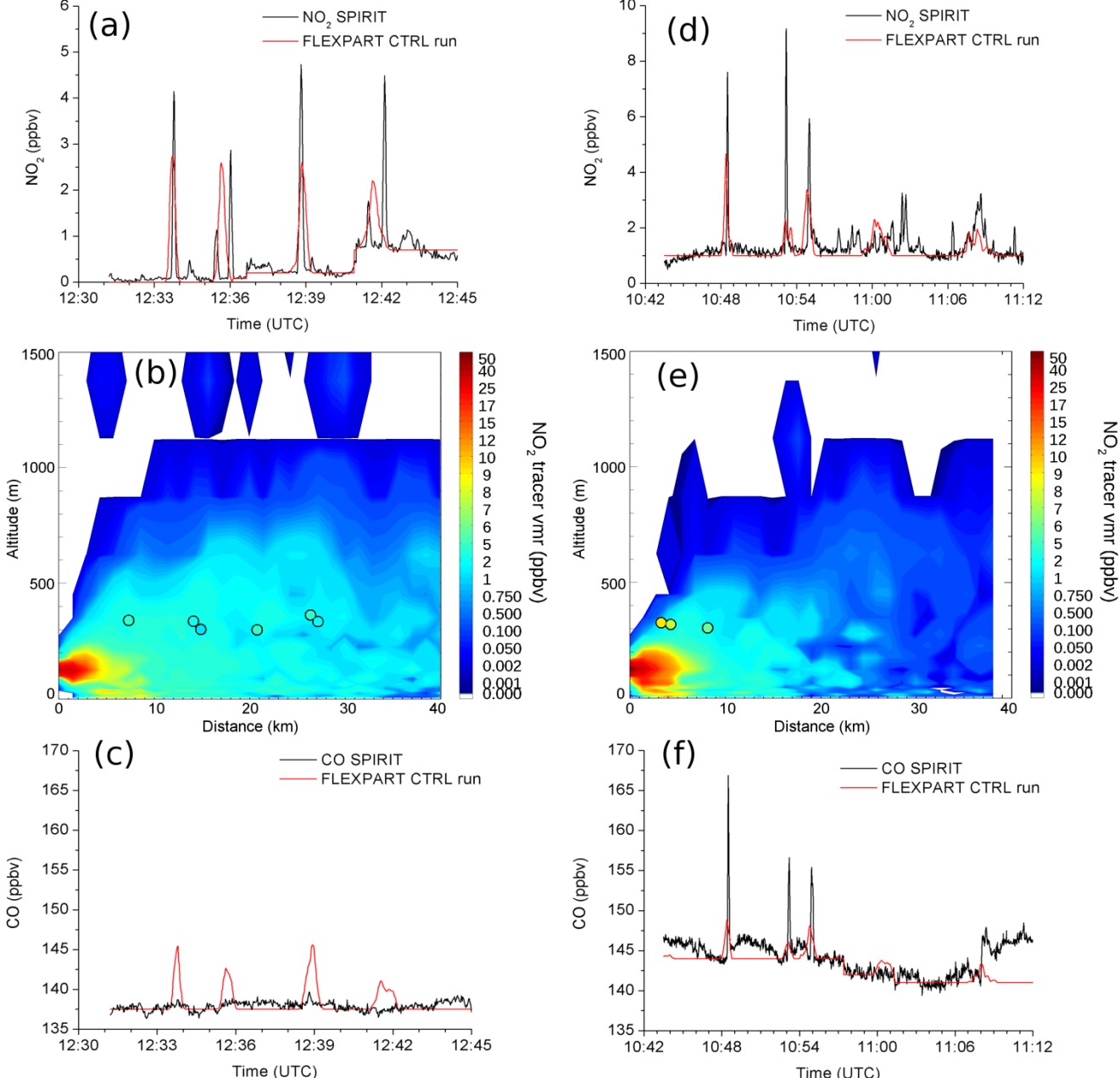

**Figure 2. Left column: July 10. (a) NO$_2$ concentration as a function of time for SPIRIT measurements (black) and FLEXPART CTRL run simulation (red). (b) Vertical section of the simulated NO$_2$ plume (CTRL run) as a function of distance from the source and altitude. The colored circles correspond to the measurement peaks. (c) CO concentration as a function of time for measurements (black) and FLEXPART CTRL run simulation (red). Right column: July 14. (d), (e) and (f) similar to (a), (b) and (c).**

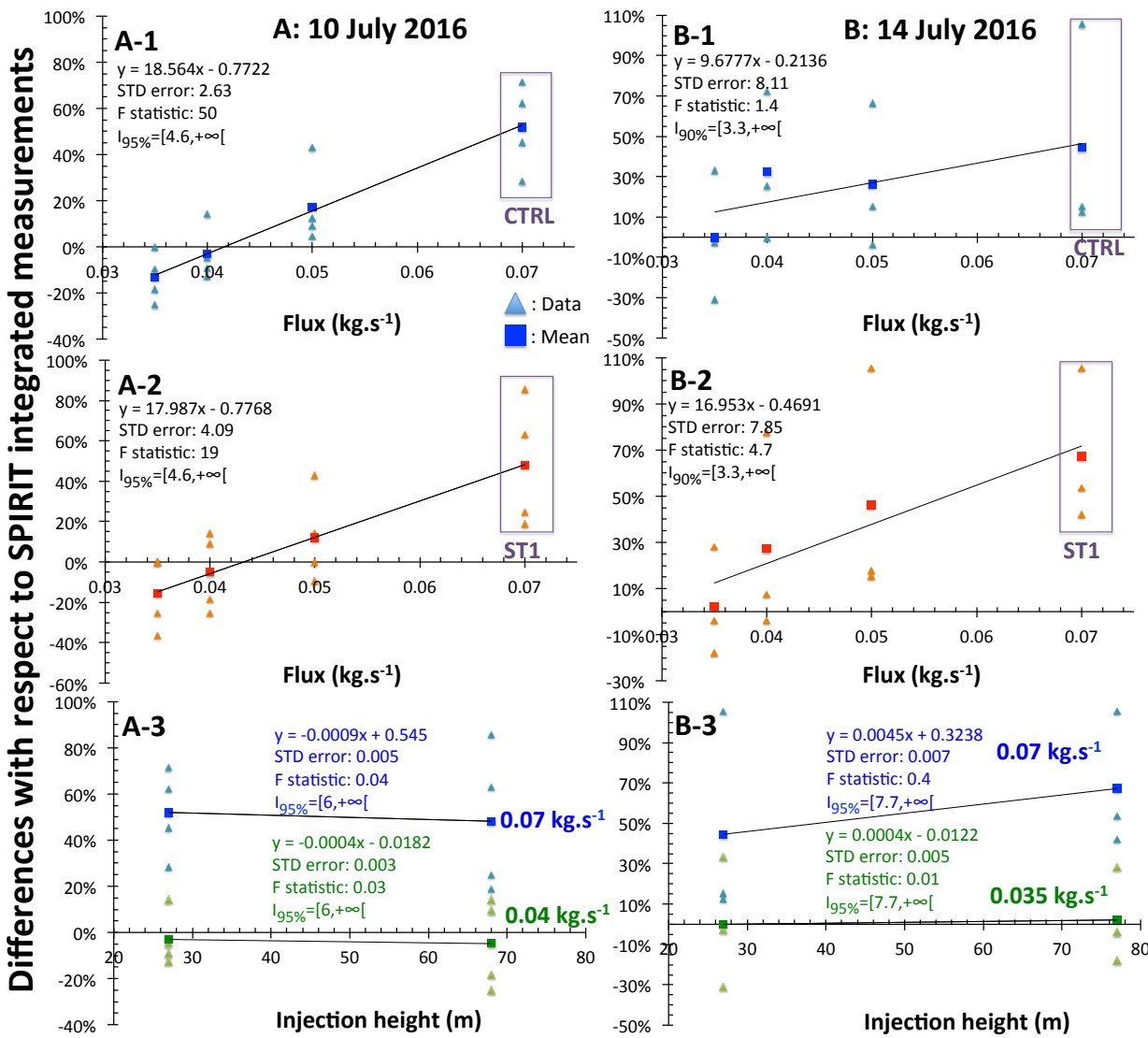

5    **Figure 3: Differences (in %) between FLEXPART simulations and SPIRIT integrated measurements depending on flux or injection height used as input in the model for A: the flight on July 10 and B: the flight on July 14. Panels A-1 and B-1 represent the change in the percentage with the flux by using the injection height from Briggs' algorithm (1965; blue data; i.e. 27 m) and Panels A-2 and B-2 with injection height from VDI 3782 (1985; orange data; i.e. 68 m (A-2) or 77 m (B-2)). Panels A-3 and B-3 represent the change in the percentage with the injection height for the flux from Deetz and Vogel (2017; blue data; 0.07 kg.s⁻¹)**

10   **and for the flux used in the sensitivity tests (green data, 0.04 kg.s⁻¹ for July 10 (A-3) and 0.035 kg.s⁻¹ for July 14 (B-3)). For all panels, triangles represent the data for all the peaks measured and squares represent the mean from these data. The slope, standard error values for the slope coefficients, the F statistic and the confidence interval ($I_{95\%}$ or $I_{90\%}$ only for panel B-1 and B-2) are added for all the plots.**

| Run name | Date of flight | NO$_2$ Flux (kg s$^{-1}$) | SO$_2$ Flux (kg s$^{-1}$) | CO Flux (kg s$^{-1}$) | Injection height (m) |
|----------|----------------|---------------------------|---------------------------|------------------------|----------------------|
| **CTRL** | 20160710 | 0.07 | 4.23×10$^{-5}$ | 0.11 | 27 |
|          | 20160714 |      |                |      |    |
| **ST1**  | 20160710 | 0.07 | Not included | Not included | 68 |
|          | 20160714 |      |              |              | 77 |
| **ST2**  | 20160710 | 0.035 - 0.05 | Not included | Not included | 27 |
|          | 20160714 |      |              |              |    |
| **ST3**  | 20160710 | 0.035 - 0.05 | Not included | Not included | 68 |
|          | 20160714 |      |              |              | 77 |

**Table 1. Flux and injection height for the reference control run (CTRL) and for the sensitivity tests (ST) for each day of flight.**