# Peer review of "Local air pollution from oil rig emissions observed during the airborne DACCIWA campaign"

_Atmospheric Chemistry and Physics, 2019_

## Referee Comment (RC1) · Joseph Pitt (Referee) · 6 Mar 2019

This paper presents aircraft measurements of pollutant emissions from an FPSO off the coast of Guinea. The case study described featured two flights within a week, with transects of the flare plume conducted at various distances downwind of the source. The Lagrangian dispersion model FLEXPART has been used in conjunction with a recent satellite-derived emissions inventory to calculate simulated plume enhancements corresponding to the aircraft sampling, which are then compared to the measurements. Sensitivity tests are performed by perturbing the magnitude of the emissions and the injection height of the plume, and it is concluded that emissions may be larger than those given by the inventory. The chemical composition of the plume, and the difference in its composition between the two flights, is also discussed.

The air quality impact of oil platforms in this region is a very important and active area of research, yet there is a lack of in situ measurements (such as those presented here) to validate satellite-based emissions estimates. Furthermore, the use of dispersion models to simulate plume enhancements for near-source aircraft sampling, as required to estimate emissions from individual facilities, has a general application for similar studies throughout the world. This study provides useful insight into both of these research areas, and I recommend its publication in ACP after the following issues are addressed.

1) The main issue I have with the study as it stands regards the way the model-measurement comparison is performed. It is not clear that comparing the peak mole fraction enhancement is the best way to do this, and at the very least this needs more discussion. For one thing, for narrow plumes the peak measured enhancement is often dependent on the e-fold time of the cell. Looking at Fig. 2 this might not be an issue here (if the actual plume width is much larger than the measurement e-fold time multiplied by the flight speed) but it is hard to be certain. Providing these details in the text might help to clear this up.

A bigger issue comes when looking at the impact of doubling the emissions inventory values. The FLEXPART simulations for this are not shown, but from the CTRL run simulation in Fig. 2 it appears that the simulated plumes are much wider than the measurements. In some cases two narrow measured plumes are combined into a single simulated plume. This feature clearly represents an error in the dispersion modelling, not the inventory. So it is possible that the inventory value is entirely correct (although obviously it's unlikely to be spot on!) but FLEXPART simply overestimates the lateral dispersion of the plume. For both flights it is concluded that one of the simulations with doubled emissions is the most representative of the measurements, but this may be a case of "right for the wrong reason" – i.e. in order to compensate for the overestimated lateral dispersion of the plume you have to bump up the emissions to get the same peak enhancements.
An alternative approach would be to compare the integrated area under each of the plume transects. This would then give a better idea of the total amount of each species within the plume (in the same way as is usually employed for calculating species-species enhancement ratios). Both approaches could be employed alongside each other as long as a suitable discussion of the issues above is included. I also think it would really help interpret what is going on here if Fig. 2 included the results from the other simulations. If you really think it's getting too cluttered then these could be moved to the supplement, but I think it's important to include them somewhere.

2) The lack of CO in the plumes measured on flight 1, and the subsequent detection of CO in the flight 2 plumes, is a really interesting result. However the discussion in Sect. 4.3 is fairly brief – it would be great to see this expanded. I'm not really clear as to what is meant by "more disturbed weather conditions" – does this mean the boundary layer was more turbulent? If so why does this mean that the combustion is less efficient? I'm not disputing that is the case, it's just not obvious to me without further explanation. I can see that probing the fluid dynamics within the flare is beyond the scope here, but are there other studies that this finding could be linked to? Were there any measurements of $CO_2$ (and ideally $CH_4$) on board? Then something quantitative could be said about the flare efficiency?

Specific points:

P2 L2 – In Nigeria all associated gas may well be flared, but in other places (at least in the UK) this associated gas is exported for use. So I think it would be more accurate to say that gas flaring is used to dispose of this natural gas in cases where the infrastructure to export it does not exist.

P3 L25 – "The concept...deeper water" – sentence reads awkwardly and needs rephrasing

P3 L28 – I suggest "dispose of" rather than "eliminate"

P3 L30 – "mixture of gas" is ambiguous – presumably this means a mixture of emitted gases?

P4 L7 – "released along time" needs rewording

P4 L11 – If I understand correctly these are just tracer particles, so their assigned mass is just a nominal quantity used in the subsequent calculations (i.e. it does not correspond to a physical mass which impacts on the particle dispersion). If so I think it would be best to clarify this, as particle mass has quite a strong association for people who work with aerosols.

P4 L20 – "for the DACCIWA project"

P5 L2 – I think it would be useful to add a sentence in here explaining both the buoyancy and momentum effects. This would make it easier to understand the subsequent assertion that the momentum effect can be ignored.

P5 L31 – P6 L3 – The description of the terms in Eq. 4 is not easy to read. It might make it clearer to define the units of the constituent terms rather than the coefficient 74.4? Also the phrase "f as the fraction of radiated heat equals to 0.27" confuses me.

P6 L30 – How is the background calculated – presumably by averaging the measured data outside the plumes? If so then it's worth stating this.

P7 L27 – "only a few quantities of pollutants" needs rewording

P8 L7 – "The turbulence increases the mixing and affects..." might be better

Fig. 1 – I trust your word that there was no SO2 measured, but you might as well add an SO2 trace to the CO plots just to demonstrate this point.

Fig. 2 – Could you make the circles around the measured data in panels b) and e) more distinct please? At least on my screen it is really hard to make these out, especially in b). See also my main point 1)

[Figure]

---

## Referee Comment (RC2) · Anonymous Referee #2 · 12 Mar 2019

This paper presents measurements from an aircraft of emissions of various air pollutants from oil rigs in the Gulf of Guinea, West Africa. The measurements are then used to quantify emissions from the rigs, which are in turn compared to calculated emissions from a new satellite derived gas-flaring emission inventory developed for the DACCIWA project. This is done using FLEXPART dispersion model simulations of the plume using the calculated emissions and then comparing the model output to the measurements. The main conclusion is that the emission rate in the inventory is too low to reproduce the measured plume concentrations using the FLEXPART model. Oil rig emissions are an important source of air pollution in this area and therefore a study like this is potentially crucial for understanding their magnitude and impact. The work is within scope of ACP however I feel there are some areas that need expanding and clarifying before it

should be published.

General points: Like reviewer 1, I am a little concerned about the way the model to measurement comparison has been done. It does seem that the peaks in the model are wider than the measured peaks and therefore comparing the maximum mixing ratio enhancement of the two could give misleading results. The authors should try comparing the integrated area under the peaks and see if this gives a different result. The effect of this should at least be discussed in the paper.

The authors also need to expand on how NO / NO2 chemistry is treated in the model. It is not clear to me whether they are changing the NO and NO2 emissions in the model to reproduce the NO2 measurements or just NO2. I would have thought most of the emission from the rig would occur as NO, with subsequent conversion to NO2 before the measurements is made. The text needs to be clearer on what chemistry is used in the model.

Does the emission from the rig include non-flaring combustion (e.g. power generation)? I would have thought that this would also be a significant source of NOx from a co-located but different source? Could this have been picked up in the measured plume but not included in the emission inventory?

It would also be good to have a short discussion as to what actual effect the oil rig emissions have on air pollution in West Africa. For instance, if the emissions are doubled in the inventory, what effect does this have on NO2 and O3 levels at the coast? I realize a full study like this is beyond the scope of this paper but some short statement should be made as to the potential impact of underestimated emissions from oil rigs in the area.

Were there measurements of CH4 made on the aircraft? If so it would have been good to see this included in the study as the rigs could also be an important CH4 source.

Specific points: P4 L27: Can the authors confirm if this is an NO2 flux or a NOx flux?

P6 L15: Is this really true. Can it really be said that because no SO2 was measured (on a relatively insensitive instrument) that no H2S was present. The authors should at least put a lower limit on the H2S that could be present.

P8 L16: this needs expanding, it is not clear what 'disturbed weather conditions' means and how this could effect the CO concentrations in the plume.

P8 L19: How will the results of the campaign improve computational flare fluid dynamics modelling?

---

## Author Comment (AC1) · 12 Jun 2019

**Manuscript title**: Local air pollution from oil rig emissions observed during the airborne DACCIWA campaign by Brocchi et al.

**RESPONSES TO JOSEPH PITT**

**We thank the reviewer for his thoughtful comments that were helpful in improving the manuscript. Changes have been made in response to his specific comments listed below (in black). Our responses appear in red, and changes in the revised manuscript are highlighted in yellow.**

1) The main issue I have with the study as it stands regards the way the model measurement comparison is performed. It is not clear that comparing the peak mole fraction enhancement is the best way to do this, and at the very least this needs more discussion. For one thing, for narrow plumes the peak measured enhancement is often dependent on the e-fold time of the cell. Looking at Fig. 2 this might not be an issue here (if the actual plume width is much larger than the measurement e-fold time multiplied by the flight speed) but it is hard to be certain. Providing these details in the text might help to clear this up.

→ Considering that the SPIRIT instrument allows measurements every 1.6 s and the e-fold time of the cell is 5.3 s, and that the average aircraft speed is 118 m s$^{-1}$ on July 10 (103 m s$^{-1}$ on July 14) during the period of the peaks of interest, even for the shortest lasting peak on each day (about 16 s on July 10 and about 19 s on July 14), the plume width is larger than the measurement e-fold time multiplied by the flight speed. We add a sentence for the worst case on July 10 (maximum aircraft speed with shortest lasting peak) in the text, p.6 lines 21-24: "Moreover, SPIRIT allows measurements every 1.6 s. Considering the case on July 10 where we have the maximum aircraft speed (118 m s$^{-1}$) and the shortest peak (lasting about 16 s), the plume width is larger than the measurement e-fold time multiplied by the flight speed. Thus, for all the narrow peaks, the maximum plume concentration is real, not a plume diluted with its surrounding environment."

An alternative approach would be to compare the integrated area under each of the plume transects. This would then give a better idea of the total amount of each species within the plume (in the same way as is usually employed for calculating species-species enhancement ratios). Both approaches could be employed alongside each other as long as a suitable discussion of the issues above is included.

→ This approach seems to be more reliable considering the dispersion modelling error in FLEXPART for example due to horizontal and vertical resolution of the windfield. However, not to complicate the reading of the paper, we decide to keep only this approach and remove the first one used by explaining why it is not possible to do a peak-to-peak comparison. The text (1$^{st}$ paragraph of section 4.2) was modified as follows:

"Concerning the second and the fourth peak (Fig. 2a), the measurements show two close peaks that FLEXPART cannot simulate individually, leading to a single and broader simulated peak. This is probably due to an error in the dispersion modelling induced by the horizontal and vertical wind field resolution that prevents us from comparing peak-to-peak concentrations. Even with a finer wind field grid mesh of 0.125°×0.125° (simulation not shown) such close peaks cannot be distinguished, suggesting a still insufficient spatial resolution.  Instead, the integrated area under each of the measured and simulated plume transects will be compared and presented in Figure 3 with the percentages representing the relative differences with respect to SPIRIT measurements…"

According to this new approach, sensitivity tests with new fluxes were performed. They are summarized in Table 1. All the results of the simulations given now correspond to the integrated area under each peak (measured and simulated). We decided to summarize the results of all these new sensitivity tests by a figure instead of a table. This is illustrated in the new figure 3.

| Run name | Date of flight | NO$_2$ Flux (kg s$^{-1}$) | SO$_2$ Flux (kg s$^{-1}$) | CO Flux (kg s$^{-1}$) | Injection height (m) |
|---|---|---|---|---|---|
| CTRL | 20160710 20160714 | 0.07 | 4.23×10$^{-5}$ kg s$^{-1}$ | 0.11 kg s$^{-1}$ | 27 |
| ST1 | 20160710 20160714 | 0.07 | Not included | Not included | 68 77 |
| ST2 | 20160710 20160714 | 0.035 - 0.05 | Not included | Not included | 27 |
| ST3 | 20160710 20160714 | 0.035 - 0.05 | Not included | Not included | 68 77 |

**Table 1. Flux and injection height for the reference control run (CTRL) and for the sensitivity tests (ST) for each day of flight.**

To interpret the results of figure 3 and show the sensitivity of FLEXPART to the input parameters (flux or injection height), simple statistical tests were made and a paragraph was added:

"To determine whether the observed linear relationship between the percentages and the flux or the injection height occurs by chance, a simple F-test is performed, assuming that the variances are homogeneous and the results follow a Gaussian distribution. F statistic coefficients are calculated and compared to the 95% or 90% confidence interval with (1, N-2) (N: total number of results) as degrees of freedom (see values in brackets [u;+∞[ in Figure 3). If the value F is included in the confidence interval then the relationship can be considered as linear. The standard errors on the slope are also added in the plots of Figure 3.

For the flight on July 10, the standard error of the slope coefficients and the F-test (95% of confidence) show linear relationships between the percentage difference and the flux. Only the results on July 14 for the plot with the injection height of VDI 3782 (1985, panel B-2) show a positive F-test but with a 90% of confidence. No conclusions can be drawn for the results on July 14 with the injection height of Briggs (1965, panel B-1). In order to show the response of FLEXPART to the injection height, panels A-3 and B-3 (Fig. 3) show the percentages versus the injection height (Briggs (1965) or VDI 3782 (1985)). FLEXPART shows similar results regardless of the injection height used as input whatever the flux used. All the cases show standard error on the slope coefficients larger than the slope itself and a F- value not included in the confidence interval (at 95% of confidence as shown in the Figure, and even 90%, not shown) . These results suggest that the differences between the two injection heights are not significant enough with respect to the vertical resolution of the model or the measurements are too far to be influenced by the changes in this parameter. However, to really

conclude about the injection height and to evaluate the flux, more measurements are needed at different altitudes and distances from the emission source. Besides the weather conditions and the functioning of the platform, the flight location is also an important parameter to be able to evaluate our measurements. Figures 2 b and e show the $NO_2$ plume simulated with FLEXPART as a function of distance from the source and altitude on July 10 and 14, respectively. The aircraft measurements, represented by the colored circles, are located at the upper part of the plume, away from the strongest concentrations. The work carried out is thus limited by the flight trajectories which were too high and too far from the FPSO platform to catch the part of the plume with the highest concentrations. The operational conditions during the flights were complex for the pilots, and safety concerns forced us to respect a minimum flight level (300 m) and a minimum distance from the source. Finally, we found that the $NO_2$ concentration difference between the measurements and the simulations does not seem to depend on the distance from the source since the measurements are already too far»

[Figure]

**Figure 3: Differences (in %) between SPIRIT integrated measurements and FLEXPART simulations depending on flux or injection height used as input in the model for A: the flight on July 10 and B: the flight on July 14 . Panels A-1 and B-1 represent the change in the percentage with the flux by using the injection height from Briggs' algorithm (1965; blue data; i.e. 27 m) and Panels A-2 and B-2 with injection height from VDI 3782 (1985; orange data; i.e. 68 m (A-2) or 77 m (B-2)). Panels A-3 and B-3 represent the change in the percentage with the injection height for the flux from Deetz and Vogel (2017; blue data; 0.07 kg.s[-1]) and for the flux used in the sensitivity tests (green data, 0.04 kg.s[-1] for July 10 (A-3) and 0.035 kg.s[-1] for July 14(B-3)). For all panels, triangles represent the data for all the peaks measured and squares represent the mean from these data. The slope, standard error values for the slope coefficients and the F statistic are added for all the plots.**

I also think it would really help interpret what is going on here if Fig. 2 included the results from the other simulations. If you really think it's getting too cluttered then these could be moved to the supplement, but I think it's important to include them somewhere.

→ We think it is better not to overload Fig. 2 as it makes it harder to read. Thus, we added a figure (Fig. S4) in the supplementary material showing the results of the sensitivity tests done with FLEXPART. We show the sensitivity tests ST2 and ST3 with the smallest differences compared to SPIRIT measurements according to Fig. 3, i.e. with a flux of 0.04 kg s$^{-1}$ on July 10 and 0.035 kg s$^{-1}$ on July 14.

[Figure]

S4. (a) July 10: NO$_2$ concentration as a function of time for SPIRIT measurements (black) and FLEXPART sensitivity test (ST) 1 (green), ST2 (purple; with a flux of 0.04 kg s$^{-1}$), ST3 (orange; with a flux of 0.04 kg s$^{-1}$). (b) Similar to (a) for July 14 with a flux of 0.035 kg s$^{-1}$ for ST2 and ST3.

2) The lack of CO in the plumes measured on flight 1, and the subsequent detection of CO in the flight 2 plumes, is a really interesting result. However the discussion in Sect. 4.3 is fairly brief – it would be great to see this expanded. I'm not really clear as to what is meant by "more disturbed weather conditions" – does this mean the boundary layer was more turbulent? If so why does this mean that the combustion is less efficient? I'm not disputing that is the case, it's just not obvious to me without further explanation. I can see that probing the fluid dynamics within the flare is beyond the scope here, but are there other studies that this finding could be linked to?

→ As far as we know, this finding could not be linked to any other study. The expression "more disturbed weather conditions" was removed and the explanations were clarified as follows: "The observed difference between July 10 and 14 in terms of CO emissions mostly lies in the different wind conditions between these two days: first, the wind speed was lower on July 14, which makes less O$_2$ available to burn with natural gas; second, it appears that the wind direction was not clearly established, as can be seen from the much more dispersed plume in Fig. 1b, resulting in incomplete combustion pockets favoring CO formation. However, a decrease in efficiency should also lead to lower temperatures and NO$_x$ emissions, which is not observed here. The results of this campaign would require to be analyzed in the light of computational fluid dynamic simulations, accounting for a realistic natural gas composition and its high-temperature chemistry, which are beyond the scope of this study."

Were there any measurements of $CO_2$ (and ideally $CH_4$) on board? Then something quantitative could be said about the flare efficiency?

→A Picarro instrument was brought to measure $CO_2$ during the campaign but it actually did not work at all because of a technical problem. $CH_4$ was measured by the SPIRIT instrument. Unfortunately, the flight conditions for these specific days were not ideal with very high temperatures inside the aircraft cabin that did not allow the $CH_4$ laser to work properly.

**Specific points:**

P2 L2 – In Nigeria all associated gas may well be flared, but in other places (at least in the UK) this associated gas is exported for use. So I think it would be more accurate to say that gas flaring is used to dispose of this natural gas in cases where the infrastructure to export it does not exist.

→ Modified following the recommendation: "Gas flaring is used to dispose of this natural gas in cases where the infrastructure to export it does not exist"

P3 L25 – "The concept...deeper water" – sentence reads awkwardly and needs rephrasing

→ It has been rephrased this way: "The concept of those platforms based on ship structure makes possible the development of small size oil fields and the exploitation of them further from the coast and thus in deeper water"

P3 L28 – I suggest "dispose of" rather than "eliminate"

→We followed the suggestion and used "dispose of".

P3 L30 – "mixture of gas" is ambiguous – presumably this means a mixture of emitted gases?

→ Yes, it has been modified as "This leads to a mixture of emitted gases".

P4 L7 – "released along time" needs rewording

→We modified it as follows: "It simulates long-range transport and dispersion of atmospheric tracers released over time".

P4 L11 – If I understand correctly these are just tracer particles, so their assigned mass is just a nominal quantity used in the subsequent calculations (i.e. it does not correspond to a physical mass which impacts on the particle dispersion). If so I think it would be best to clarify this, as particle mass has quite a strong association for people who work with aerosols.

→ We agree, the model only considered $NO_2$ as tracer particles. All the particles undergo the same transport. The mass input in the "release" file is considered as a flux of $NO_2$ (a mass release during time). However, the other parameters such as molecular mass, OH reaction rate constant, Henry law constant… have also to be included and are taken into account for the wet and dry deposition calculations. We removed the sentence: "In this mode, each particle is associated to a given mass of tracer released during the time of the simulation" and add complementary information in the text and in Table 1: "The particles are released with the chemical properties of $NO_2$, CO and $SO_2$ using constant emissions from Deetz and Vogel (2017) inventory during 7 h with a spin-up of 5 h, allowing the model to be balanced independently from the initial conditions. During the simulation, the $NO_2$ and $SO_2$ like particles mass is lost by wet and dry deposition and by OH reaction (concentrations from GEOS-CHEM model; Technical note FLEXPART v8.2, http://flexpart.eu/downloads/26), which allows a lifetime of about 3 h at 298 K in the MBL for $NO_2$. CO like particles mass is only lost by OH reaction."

P4 L20 – "for the DACCIWA project"

→Modified following the suggestion.

P5 L2 – I think it would be useful to add a sentence in here explaining both the buoyancy and momentum effects. This would make it easier to understand the subsequent assertion that the momentum effect can be ignored.

→We added a small definition for each effect in the text: "Indeed, the buoyancy corresponds to a density ratio between the air parcel and its colder surrounding environment and leads to the rise of this parcel under the influence of gravity. This effect is to be distinguished from the momentum effect defined as the product of an element mass by its velocity, which can be neglected for such high temperature plume (Briggs, 1965)."

P5 L31 – P6 L3 – The description of the terms in Eq. 4 is not easy to read. It might make it clearer to define the units of the constituent terms rather than the coefficient 74.4? Also the phrase "f as the fraction of radiated heat equals to 0.27" confuses me.

→ The units of all constituents, in particular Q and H (in MW), have been added in the manuscript.

→ The phrase has been rewritten as follows: "f (as) the fraction of radiated heat set to 0.27 by Deetz and Vogel (2017), after having averaged the f values given in Guigard et al. (2000)".

P6 L30 – How is the background calculated – presumably by averaging the measured data outside the plumes? If so then it's worth stating this.

→ It is true that the background has been calculated by averaging the measured data outside the plumes. We added a sentence in the text explaining it: "This value is an average of the measurements taken outside the plume".

P7 L27 – "only a few quantities of pollutants" needs rewording

→It has been modified as follows: "only a small fraction of pollutants".

P8 L7 – "The turbulence increases the mixing and affects..." might be better

→ Yes, modified accordingly

Fig. 1 – I trust your word that there was no $SO_2$ measured, but you might as well add an SO2 trace to the CO plots just to demonstrate this point.

→ More precisely, we wrote that no $SO_2$ (and CO) peaks were detected simultaneously to $NO_2$ peaks. This is shown in the plots we added for the $SO_2$ measurements in Figure 1. The legend was also modified. $SO_2$ FLEXPART simulation was also realized and shows insignificant $SO_2$ concentration at the aircraft sampled location. A sentence has been added in the text: "Considering the very low $SO_2$ flux, the FLEXPART simulations in the CTRL run induce insignificant $SO_2$ concentration at aircraft sampled location (results not shown)."

[Figure]

Figure 1: (a) NO$_2$ concentration as a function of the flight trajectory downwind of the FPSO plume for July 10. The black arrows show the wind direction (from ECMWF). (c) NO$_2$, aerosol, CO and SO$_2$ concentrations as a function of time zoomed in a part of the flight trajectory in (a). The peaks studied are labelled by a number (from 1 to 4). (b) and (d) are similar to (a) and (c) for July 14.

Fig. 2 – Could you make the circles around the measured data in panels b) and e) more distinct please? At least on my screen it is really hard to make these out, especially in b). See also my main point 1)

→The circles on Figure 2 have been highlighted as shown below. Moreover, a small error was detected on this figure. Fig. 2 (e) was not taken at the right time in the first version of the paper. This is modified in this version but it does not modify the conclusions of the article.

---

## Author Comment (AC2) · 12 Jun 2019

**Manuscript title**: Local air pollution from oil rig emissions observed during the airborne DACCIWA campaign by Brocchi et al.

**RESPONSES TO ANONYMOUS REFEREE #2**

**We thank the reviewer for his relevant comments that were helpful in improving the manuscript. Changes have been made in response to his specific comments listed below (in black). Our responses appear in red, and changes in the revised manuscript are highlighted in yellow.**

General points:

Like reviewer 1, I am a little concerned about the way the model to measurement comparison has been done. It does seem that the peaks in the model are wider than the measured peaks and therefore comparing the maximum mixing ratio enhancement of the two could give misleading results. The authors should try comparing the integrated area under the peaks and see if this gives a different result. The effect of this should at least be discussed in the paper.

→ We agree on this point. An error in the dispersion modelling has been mentioned by Referee #1 and leads to simulated peaks wider than the measured ones. We used the approach that consists in comparing the integrated areas and modified the text ($2^{nd}$ paragraph of section 4.2) as follows:

"Concerning the second and the fourth peak (Fig. 2a), the measurements show two close peaks that FLEXPART cannot simulate individually, leading to a single and broader simulated peak. This is probably due to an error in the dispersion modelling induced by the horizontal and vertical wind field resolution that prevents us from comparing peak-to-peak concentrations. Even with a finer wind field grid mesh of 0.125°×0.125° (simulation not shown) such close peaks cannot be distinguished, suggesting a still insufficient spatial resolution. Instead, the integrated area under each of the measured and simulated plume transects will be compared and presented in Figure 3 with the percentages representing the relative differences with respect to SPIRIT measurements."

According to this new approach, sensitivity tests with new fluxes were performed. They are summarized in Table 1. All the results of the simulations given now correspond to the integrated area under each peak (measured and simulated). We decided to summarize the results of all these new sensitivity tests by a figure instead of a table. This is illustrated in Figure 3 and results are discussed from the $2^{nd}$ paragraph of section 4.2 onwards.

[Figure]

**Figure 3: Differences (in %) between SPIRIT integrated measurements and FLEXPART simulations depending on flux or injection height used as input in the model for A: the flight on July 10 and B: the flight on July 14 . Panels A-1 and B-1 represent the change in the percentage with the flux by using the injection height from Briggs' algorithm (1965; blue data; i.e. 27 m) and Panels A-2 and B-2 with injection height from VDI 3782 (1985; orange data; i.e. 68 m (A-2) or 77 m (B-2)). Panels A-3 and B-3 represent the change in the percentage with the injection height for the flux from Deetz and Vogel (2017; blue data; 0.07 kg.s⁻¹) and for the flux used in the sensitivity tests (green data, 0.04 kg.s⁻¹ for July 10 (A-3) and 0.035 kg.s⁻¹ for July 14(B-3)). For all panels, triangles represent the data for all the peaks measured and squares represent the mean from these data. The slope, standard error values for the slope coefficients and the F statistic are added for all the plots.**

The authors also need to expand on how NO / $NO_2$ chemistry is treated in the model. It is not clear to me whether they are changing the NO and $NO_2$ emissions in the model to reproduce the $NO_2$ measurements or just $NO_2$. I would have thought most of the emission from the rig would occur as NO, with subsequent conversion to $NO_2$ before the measurements is made. The text needs to be clearer on what chemistry is used in the model.

→ FLEXPART model uses a very simple chemistry: the particles are released with chemical properties like $NO_2$ using constant emissions (mass release during time). During the simulation, the $NO_2$ like particles mass is lost by wet and dry deposition and by OH reaction, only.

The emissions used to initialize FLEXPART come from Deetz and Vogel (2017) inventory and are expressed in terms of $NO_2$. They considered a rapid conversion for a part of the NO emissions into $NO_2$ very close to the source. This is confirmed with some quick simulations with a box model

(AtChem/MCM) showing conversion in the first 5 minutes of the simulation. Most of the atmospheric models consider $NO_2$ as primary emissions while it is in fact a secondary emission. A sentence has been included in Section 3.2:

"Fourth, for such a temperature, $NO_2$ is considered as a primary pollutant coming from the rapid conversion of NO close to the source, and the inventory does not include any later transformation of the species"

Does the emission from the rig include non-flaring combustion (e.g. power generation)?

I would have thought that this would also be a significant source of NOx from a collocated but different source? Could this have been picked up in the measured plume but not included in the emission inventory?

→ The emission inventory (Deetz and Vogel) from the rig includes only flaring emissions. Emissions from power generation for the facility could be also a source of NOx. However, considering that the aircraft was flying at an altitude of about 300 m, only the emissions that have been heated up at very high temperature could reach rapidly and directly this altitude and induce a very localized spike in the $NO_2$ signal. This is why we think that we have only measured flaring emissions.

It would also be good to have a short discussion as to what actual effect the oil rig emissions have on air pollution in West Africa. For instance, if the emissions are doubled in the inventory, what effect does this have on NO2 and O3 levels at the coast? I realize a full study like this is beyond the scope of this paper but some short statement should be made as to the potential impact of underestimated emissions from oil rigs in the area.

→ The impact of the FPSO emissions on the air quality on the coast is not mentioned in the paper. In fact, the platform is situated 70 km downwind of the coast, which supposes that there is no impact of direct emissions of $NO_2$. However, we agree that $O_3$ and other pollutants such as $CO_2$, $CH_4$, BC and CO with relatively long lifetime should impact the air quality on the coast. As you mentioned, "a full study like this is beyond the scope of this paper" but as suggested a short statement is included in the paper with more information directly included in the text (end of section 4.2) and in the conclusion:

"We can use the best simulation on each day to estimate the percentage of pollutants transported inside and above the MBL. In both cases, about 90% of the pollutants stay inside the MBL and are susceptible to impact the population living along the coastline. Measurements made along the coastline have shown that $NO_2$ concentrations are generally greater than 2 ppb for suburban sites and greater than 20 ppb near industrial sites (Bahino et al. 2018). Given the wind velocity (from 6.6 to 9.4 m s$^{-1}$), the air masses attain the coast in 2 to 3 hours, which does not allow to bring significant $NO_2$ concentrations to impact air quality in this area. This is confirmed looking at FLEXPART simulations in Fig. 2b and 2e. They show that $NO_2$ concentrations are already very low (< 1 ppbv) from 40 km from the source on July 10, and even closer (from 20 km from the source) on July 14. The distance between the coast and the emission source following the wind direction being 70 km, only pollutants with a relatively long lifetime or secondary pollutant as $O_3$ can impact the air quality of the coast."

"An estimation of the pollutant distribution above or inside the MBL shows that the pollutants stay mainly inside the MBL, limiting the transport to the coastline located 70 km downwind of the FPSO.

Were there measurements of CH4 made on the aircraft? If so it would have been good to see this included in the study as the rigs could also be an important CH4 source.

→ The SPIRIT instrument measured $CH_4$ during the campaign. Unfortunately, the flight conditions for these specific days were not ideal with very high temperatures inside the aircraft cabin (low altitude make less efficient the air conditioning) that did not allow the $CH_4$ laser to work properly.

Specific points:

P4 L27: Can the authors confirm if this is an NO2 flux or a NOx flux?

→ We confirm that this is an $NO_2$ flux given in the inventory. A NO flux is also given but no direct $NO_x$ flux is provided.

P6 L15: Is this really true. Can it really be said that because no SO2 was measured (on a relatively insensitive instrument) that no H2S was present. The authors should at least put a lower limit on the H2S that could be present.

→ It is well known the oxidation of $H_2S$ fully leads to $SO_2$ formation. See any general book about Atmospheric Chemistry; we added the reference (Sonibare and Akeredolu, 2004) already quoted in the present paper. Sentence is added in the paper section 4.1:

"Knowing that SO2 comes from H2S combustion (Sonibare and Akeredolu, 2004), these results suggest that a gas composition of 0.03% of H2S induces an emission of SO2 concentration lower than the detection limit of the instrument from 3 km of our measurements or the natural gas composition given by Deetz and Vogel (2017) for the Niger Delta is different from that in Ghana for those two flights."

P8 L16: this needs expanding, it is not clear what 'disturbed weather conditions' means and how this could effect the CO concentrations in the plume.

We agree this expression was rather confusing so we replaced it by a more specific one which should clarify the reason why CO formation is increased: "the wind direction was not clearly established, as can be seen from the much more dispersed plume in Fig. 1b, resulting in incomplete combustion pockets favoring CO formation."

P8 L19: How will the results of the campaign improve computational flare fluid dynamics modelling?

→ This sentence was deleted

---

## Author Response (AR2)

**Manuscript title**: Local air pollution from oil rig emissions observed during the airborne DACCIWA campaign by Brocchi et al.

**RESPONSES TO ANONYMOUS REFEREE #2**

We thank the reviewer for this relevant comment.

**The authors have done a very nice job revising the manuscript after the initial reviewer comments. I still feel more discussion should be given as to the non-flaring combustion as a source of NOx from the rig. I am still not totally convinced that only flaring emissions would be seen at an altitude of 300m so perhaps the authors could go in to some more detail about the likely magnitude of non-flaring NOx emissions?**

A paragraph is added in the page 8 lines 16:

« Emissions from oil and gas production vary depending on the operating conditions of the platforms making them variable and hard to analyse (Law et al., 2017). Considering this, it is possible to have sources of NOx (and $SO_2$ if present) from non-flaring combustion processes like the power generation for the facility (Villasenor et al., 2003). No review can be found in the literature to estimate the magnitude of those emissions. As a comparison, during the ACCESS aircraft campaign (Tuccella et al., 2017) in the Arctic, the maximum NOx mixing ratio associated to oil and gas platforms is about 10 ppbv (Fig. 5 in Tuccella et al (2017)). For the ACCESS campaign, it is known that the facilities were operating under normal conditions and the flight altitude ranges from 120-250 m. The comparison of the FPSO in the Gulf of Guinea with the platforms in Arctic shows that the $NO_2$ measured during DACCIWA campaign is in the range of a normal functioning mode. As mentioned in Zhang et al (2019), the sources associated to non-flaring combustion are not always negligible, depending on the operating conditions. Thus, we cannot exclude a contribution from another NOx source overestimating the results, but this contribution is not important when gas flaring activity is in normal mode (Zhang et al., 2019). »